# Characterization and Diversity of *Klebsiella pneumoniae* Prophages

**DOI:** 10.3390/ijms24119116

**Published:** 2023-05-23

**Authors:** Fuqiang Kang, Zili Chai, Beiping Li, Mingda Hu, Zilong Yang, Xia Wang, Wenting Liu, Hongguang Ren, Yuan Jin, Junjie Yue

**Affiliations:** Laboratory of Advanced Biotechnology & State Key Laboratory of Pathogen and Biosecurity, Beijing Institute of Biotechnology, Beijing 100071, China; kfq960605@163.com (F.K.);

**Keywords:** *Klebsiella pneumoniae*, prophage, GC content, virulence factor, antimicrobial resistance

## Abstract

*Klebsiella pneumoniae* is a common human commensal and opportunistic pathogen. In recent years, the clinical isolation and resistance rates of *K. pneumoniae* have shown a yearly increase, leading to a special interest in mobile genetic elements. Prophages are a representative class of mobile genetic elements that can carry host-friendly genes, transfer horizontally between strains, and coevolve with the host’s genome. In this study, we identified 15,946 prophages from the genomes of 1437 fully assembled *K. pneumoniae* deposited in the NCBI database, with 9755 prophages on chromosomes and 6191 prophages on plasmids. We found prophages to be notably diverse and widely disseminated in the *K. pneumoniae* genomes. The *K. pneumoniae* prophages encoded multiple putative virulence factors and antibiotic resistance genes. The comparison of strain types with prophage types suggests that the two may be related. The differences in GC content between the same type of prophages and the genomic region in which they were located indicates the alien properties of the prophages. The overall distribution of GC content suggests that prophages integrated on chromosomes and plasmids may have different evolutionary characteristics. These results suggest a high prevalence of prophages in the *K. pneumoniae* genome and highlight the effect of prophages on strain characterization.

## 1. Introduction

*Klebsiella pneumoniae*, a member of the Enterobacteriaceae family, is a Gram-negative bacterium responsible for a wide range of community-acquired, health-care-associated, and nosocomial infections, such as pneumonia, bacteremia, and urinary tract infections, and is associated with high morbidity and mortality rates [1]. *K. pneumoniae* has been found to colonize multiple body sites in humans, including the respiratory tract, intestines, and skin. It is also associated with a wide variety of animals and plants and is found in water, soil, and grains. *K. pneumoniae* is considered one of the most important opportunistic pathogens in humans, with most infections occurring in newborns, elderly individuals, and immunocompromised populations. In recent years, multidrug resistant (MDR) and hypervirulent *K. pneumoniae* (hvKP) have been on the rise, leading to increasingly limited treatment options. The emergence of MDR hvKP has caused concern in many countries around the world.

Temperate (lysogenic) phages are viruses that follow an alternative life cycle, integrating their genome into the host and becoming prophages. In this state, the phage DNA replicates along with the host cell (lysogen) and is maintained in the bacterial population. Microbial genomics sequence analysis has revealed that substantial amounts of bacterial DNA are, in fact, prophage-like elements and prophage remnants in some bacteria [2,3], suggesting that such mobile genetic elements are widely present in bacteria. This verifies the outstanding ability of prophage to readily excise from their host genome, enter the lytic cycle, and infect other cells, thereby spreading their genes to other bacteria. It has also become increasingly clear that prophage DNA plays an important role in the evolution of bacterial pathogenicity [4]. This also helps us to better understand that the interaction between phage and bacteria is not a simple parasite–host relationship but a two-way coevolution of virus and bacterial genomes.

Understanding prophages is important not only for the exploration of high genomic variability in *K. pneumoniae* but also for an overall genomics perspective on the virulences and drug resistances of bacterial strains. This study sought to understand the impact of prophage integration on the genome structure and characteristics of *K. pneumoniae*. In addition, this study provides information on the distribution and prevalence of various types of prophages in the genome of *K. pneumoniae* and compares the differences in prophage types and properties on chromosomes and plasmids. Through the comparison of GC content, it is suggested that the chromosomes and plasmids of *K. pneumoniae* may adopt different evolutionary ways.

## 2. Results

### 2.1. Prophages Are Abundant in K. pneumoniae and Vary between Chromosomes and Plasmids

A total of 15,946 prophages were detected, of which 9755 prophages were on chromosomes and 6191 prophages were on plasmids (Appendix A). We found prophages present in each strain, and the total number of prophages per strain ranged from 1 to 25. A total of 1433 strains were found to have integrated at least two prophages, and most strains harbored either eight (n = 137) or seven (n = 125) prophages, while the remaining four strains had only one prophage, with an average of approximately 11.10 prophages per strain (Figure 1), which indicated that prophages were abundant in the *K. pneumoniae* genome.

Prophages were detected in all chromosome sequences of *K. pneumoniae*, and no putative prophage was identified in 38.14% of all plasmids. Of the prophages identified in this study, 6224 were classified as intact, 2981 as questionable, and 6741 as incomplete. There were differences in the distribution of different completeness prophages on the chromosomes and plasmids of *K. pneumoniae*. Intact prophages were found in the chromosomes of 1399 out of 1437 strains. The intact prophages in chromosomes were about 50.84% of all prophages, and the rest classified as questionable (17.52%) or incomplete (31.64%) prophages (Figure 2A). On plasmids, intact, questionable, and incomplete prophages accounted for 20.41%, 20.56%, and 59.03% of all prophages, respectively (Figure 2B). The presence of “intact prophages” in the genome of strains usually represents the bacteria newly infected with integrated phages [5], and defective prophages (a collective term for “questionable prophages” and “incomplete prophages”) may be attributed to the inactivation of prophages by mutations or gene loss under strong stress selection [6].

As mentioned above, approximately 38.14% of the plasmids in *K. pneumoniae* did not contain prophages, and the plasmids of *K. pneumoniae* ranged considerably in terms of length [7]. We wanted to know whether the plasmid length of *K. pneumoniae* was related to the presence of prophages, and, based on this, we compared the length of the plasmid and the number of prophages it carried. The results showed that the other plasmids harbored at least one prophage, and the number of prophages in some plasmids was even as high as eight, except for plasmids without prophages. The plasmids that harbored prophages exhibited a considerable increase in length compared with those without prophages (Kruskal–Wallis, df = 2, *p* = 0, Figure 2C, Appendix A). Plasmids with one prophage were significantly longer than plasmids without prophages (Kruskal–Wallis, df = 1, *p* = 0, Figure 2C, Appendix A). Plasmids with two or more prophages were even greater in length than those with only one prophage (Kruskal–Wallis, df = 1, *p* < 0.001, Figure 2C, Appendix A). This implied that prophage integration was associated with the plasmid length of *K. pneumoniae*. Likewise, the number of prophages per chromosome was found to be positively correlated with chromosome length (Spearman’s rho = 0.59, *p* < 0.0001, Appendix A). A trend that could be seen was, as the number of prophages increased, the chromosomes gradually increased in size.

### 2.2. The Types of Prophages Integrated on Chromosomes and Plasmids Are Significantly Different

We investigated the prophage types integrated into the genome in each strain. The name of each prophage type was described using the most common phage indicated by the PHASTER database. We obtained a total of 332 different prophage types on chromosomes and plasmids, of which 239 appeared on chromosomes and 173 on plasmids. Differences in the integration types of prophages between the chromosomes and plasmids in *K. pneumoniae* were then compared. We observed that the types of prophages were significantly different in the top ten prophages of chromosomes and plasmids. Except for Klebsi_phiKO2, no other type of prophage was found to co-occur in the top ten prophages for the chromosome and plasmid integration (Figure 3A). Prophage Escher_RCS47 was the most prophage type found on the plasmids. Although Escher_RCS47 was also more abundant on the chromosome, its integration ratio ranked 11th. The top ten prophage types with a large proportion of integration on the plasmid were also integrated on the chromosome, but their respective proportions of integration on the chromosomes were less than 1% (except Escher_RCS47 and Klebsi_phiKO2, with the proportions of integrations on the chromosome of 3.64% and 3.90%, respectively). In the top ten prophage types with more chromosome integrations, it was found that their integration ratios on plasmids were also less than 1% (except Klebsi_phiKO2, which was 2.18%) and even less than 0.5%. Three kinds of prophages were not found on the plasmids, Entero_P4, Klebsi_ST512_KPC3phi13.2, and Klebsi_ST437_OXA245phi4.1, and their proportions on chromosomes were 10.31%, 4.46%, and 3.73%, respectively. It is worth noting that prophage Entero_P4 was the most integrated prophage type on the chromosome. Overall, our results showed that the integration of the top ten prophage types with the highest integration ratios in chromosomes and plasmids was significantly different, and their integration advantages were related to the genomic regions of the strains.

In the sequence set consisting of 1437 strains of *K. pneumoniae* in this study, 1412 strains had known sequence types, which were divided into 263 known, different MLST types, and the other 25 strains had new sequence types. A total of 107 K loci and 11 O loci were identified in 1437 strains. In the sequence typing of *K. pneumoniae* in this study, the number of ST11 strains was the highest (290/1412, 20.54%). Among all K-locus types, strains of the KL64 type had the highest percentage (224/1437, 15.59%). Among all O-locus types, strains of the O1/O2v1 and O1/O2v2 types had the highest percentages (507/1437, 35.28% and 504/1437, 35.07%, respectively).

To understand the relationships between prophage types and strain types, we selected the top eighty percent of prophage types that appeared on the chromosome, the top seventy percent of known sequence types, the top eighty percent of K-locus types, and all O-locus types. Then, the expression of the MLST-, K-, and O-locus types of strains and their respective chromosomally integrated prophage types were linked. Entero_P4 was the prophage type with the highest percentage of chromosomal integration, mainly in *K. pneumoniae* types ST11 and ST258. In addition to this prophage, ST11-type strains had a higher number of chromosomes integrated with Escher_HK639, Escher_phiV10, Klebsi_ST437_OXA245phi4.1, Klebsi_ST512_KPC3phi13.2, and Salmon_SEN34 (Figure 3B, Appendix A), and these types of prophages also had high percentages of integration on chromosomes (Figure 3A). In other sequence types, the distribution of these prophages was rare or even absent. Although it can be seen that some types of prophages appeared more frequently in ST11-type strains, there were several types of prophages that did not appear on the chromosomes of ST11-type strains, such as Klebsi_ST16_OXA48phi5.4, Entero_phiP27, and Entero_HK140. Escher_500465_1 was the second most abundant prophage on the chromosomes, distributed in a variety of strains of different sequence types. However, the number of such prophages in each sequence type was small, all less than 50 (Figure 3B, Appendix A). In the K types of *K. pneumoniae*, the prophage Entero_P4 was mainly located on the chromosomes of the KL64, KL47, and KL107 strains. Most types of prophages were also predominantly concentrated in these three K-locus types. Consistent with the results presented by the sequence type, the prophage Escher_500465_1 also occurred in several different K-locus types. However, this type of prophage was more distributed in KL2, KL51, and KL102 than in the three aforementioned K-locus types (Figure 3C, Appendix A). Among all O types, O1/O2v1 and O1/O2v2 type strains integrated the most abundant types and numbers of prophages. The prophages Entero_P4 and Escher_500465_1 were the prophage types with the highest percentages of integration in these two O subtypes. A centralized distribution of some prophages was also observed in strains of the OL101 subtype (Figure 3D, Appendix A). This suggested that strain types may be related to the types of prophages integrated on the chromosome.

To understand whether there are differences in the size of prophages located on chromosomes and plasmids, all prophages were classified and compared according to their position in the genome of the strain and their degree of completeness. Intact prophages were our first focus, as they were most likely to be structurally and functionally intact. The results showed that the length of intact prophages found in chromosome sequences was significantly longer than that of prophages on plasmids (Mann–Whitney, *p* < 0.001, Figure 4, Appendix A). Furthermore, prophages of questionable and incomplete types on chromosomes also had larger genomes than prophages of the same type on plasmids (Mann–Whitney, *p* < 0.001, *p* = 0.885, respectively, Appendix A). Our results suggested that prophages may be selective for different genetic regions in the genome of the strain, in addition to selecting specific integration sites [8,9] during their entry into the genome of the host bacterium.

### 2.3. Prophages on Chromosomes and Plasmids Have Different Effects on the Virulence and Drug Resistance of Strains

Prophages provide genomic plasticity and host adaptation for their bacteria [10] and act as important vehicles carrying virulence factor (VF) [11] and antibiotic resistance (AMR) genes [12]. A large number of virulence factors and antibiotic resistance genes were also identified in the prophages in this study. To understand the impact of prophages on virulence and drug resistance in their host strains, we assessed the proportion of VF and AMR genes encoded by prophages on chromosomes and plasmids of *K. pneumoniae* as a percentage of the whole strain. The analysis revealed some prophages that encoded neither VF nor AMR genes (Figure 5A,B), and these prophages may have other functions for the host strain. In terms of virulence factors, the prophage on the chromosome contributed more to the virulence of the whole strain. As the number of prophages increased, the median virulence ratio also increased. In some strains, chromosomal prophages encoded more than 10% of VF genes. In contrast, most of the plasmid-encoded VF genes were concentrated in the range of 0–2% of the strain, with a maximum of no more than 5%. Moreover, the proportion of VF genes they encoded hardly increased with the number of prophages (Figure 5A). In terms of antibiotic resistance genes, as the number of prophages increased, the proportion of AMR genes encoded by prophages on chromosomes and plasmids increased as a percentage of the AMR genes in their strains. The median resistance rate of prophages on chromosomes for the whole strain generally did not exceed 5%, while the majority of the median resistance rates for prophages on plasmids exceeded 5% and approached 10% in some strains, with the maximum resistance rate approaching 25%. These results implied that prophages on plasmids may have a greater impact on strain drug resistance than those on chromosomes (Figure 5B). Overall, the differences in proportions for each strain indicated that the ratio between VF and AMR genes differed greatly depending on the prophage integrated by the strain. This also suggested that prophages are indeed one of the more important classes of mobile genetic elements contributing to virulence and antibiotic resistance.

### 2.4. The Classes and Proportions of VF and AMR Genes Encoded by Prophages on Chromosomes and Plasmids Are Different

At present, some studies on VF and AMR genes encoded by prophages have primarily focused on intact prophages [13], while the effect of questionable and incomplete types on host bacterial virulence and resistance seems to be less well studied. In fact, even degraded questionable or incomplete prophages can provide many benefits for the host to cope with adverse environmental conditions [14]. In this study, we found that the proportion of questionable and incomplete prophages carrying VF or AMR genes was greater than that of intact prophages carrying the same type of genes, except for those prophages that may not contain any VF and AMR genes (Figure 5C,D). This shows that questionable and incomplete prophages, especially incomplete prophages, seem to play a greater role in spreading VF and AMR genes. This suggests that VF gene- or AMR gene-encoding prophages are prone to inactivation and degradation and thus are likely to become incomplete prophages.

Some of the prophages of *K. pneumoniae* have previously been found to carry AMR genes [15]. It has also been shown that prophages do not act as vectors for the transfer of AMR genes in strains [16]. In the present study, genes capable of encoding VF or AMR were found in both chromosome- and plasmid-integrated prophages. On the chromosome, approximately 69% of prophages did not contain any VF and AMR genes, with intact prophages predominating at approximately 42%. The proportion of prophages carrying only-VF and only-AMR genes was close, approximately 11%. At the same time, we also identified prophages that likely had both virulence factors and drug resistance genes, approximately 9%. A higher proportion of prophages carried VF genes on chromosomes compared to plasmids (Figure 5C, Appendix A). Approximately 56% of all prophages on plasmids may not encode any VF and AMR genes, with incomplete prophages predominating at approximately 38%. There were significantly more prophages carrying only AMR genes than prophages carrying only VFs, regardless of whether their prophages were intact, questionable, or incomplete. The proportion of prophages with both VF and AMR genes was low compared to chromosomes, accounting for less than 3% of all prophages on plasmids. Unlike chromosomes, however, most prophages on plasmids carried genes associated with drug resistance (Figure 5D, Appendix A).

We further reviewed the functions of the VF genes encoded by the prophages, and the results showed that these virulence genes are associated with antimicrobial activity or competitive advantage, biofilm, effector delivery system, adherence, exotoxin, immune modulation, regulation, stress survival, nutritional or metabolic factor, exoenzyme, and invasion. Among these aforementioned virulence factor functions, antimicrobial activity or competitive advantage occurs only in chromosome-situated prophages, while exoenzyme and invasion occur only in prophages located on plasmids. Furthermore, the number of prophages with these virulence-related functions varied among chromosome- and plasmid-integrated prophages (Figure 5E, Appendix A). In terms of resistance mechanisms of antibiotic resistance genes, prophages encoding resistance genes on both chromosomes and plasmids had six identical resistance mechanisms, namely, antibiotic efflux, antibiotic inactivation, antibiotic target alteration, antibiotic target protection, antibiotic target replacement, and reduced permeability to antibiotics. In addition, prophages with antibiotic resistance on the chromosome additionally have a mechanism of resistance by absence. These prophages with different resistance mechanisms have their own advantages of chromosomes and plasmids. The resistance mechanisms of prophages on chromosomes were mainly antibiotic target alterations and antibiotic effluxes, and the resistance mechanisms of plasmid-integrated prophages were mainly antibiotic inactivations (Figure 5F, Appendix A).

### 2.5. The Integration Characteristics of Prophages on Chromosomes and Plasmids Are Likely Different

As the nucleotide composition is variable across species, we can understand the evolutionary characteristics of a genetic fragment by comparing whether its guanine–cytosine content (GC%) differs from the genome in which it was found. The newly acquired genetic fragments differ in GC content from the genomic region in which they were located, which provides one of the markers to reveal whether they originated from other species due to horizontal gene transfer (HGT). If the GC content of a foreign fragment is similar to the GC content of the genomic region in which it was located, it indicates that it originated from an earlier integration. In contrast, if the nucleotide composition of a foreign fragment is significantly different from its localization, it indicates that the genetic fragment may have originated from a recent integration.

To understand the integration characteristics of the prophage on chromosomes and plasmids, CodonW was used to calculate the GC content of the prophage with the genetic region of the genome in which it was located (Appendix A), and then the GC content of the two was compared. Prophages Entero_P4 and Escher_RCS47 were the most common types of prophages for chromosomes and plasmids, respectively (Figure 3A). Therefore, we first compared whether the GC content of these two prophages differed from the genomic region in which they were located. The average GC% of all 1006 Entero_P4 prophages was 53.7%, while the average bacterial chromosome GC% was 58.5% (Figure 6A, Appendix A). It has been assumed that prophage Entero_P4 in *K. pneumoniae* was transferred via HGT, and it occurred in a recent integration, as the G+C content of prophage Entero_P4 was largely different from the genomic G+C content of this organism. The average GC% of all 355 Escher_RCS47 prophages integrated on the chromosome was 52.2%, while the average bacterial chromosome GC% was 58.5% (Figure 6B, Appendix A). The average GC% of all 2516 Escher_RCS47 prophages integrated on plasmid was 54.4%, while the average bacterial plasmid GC% was 53.1% (Figure 6C, Appendix A). The comparison of GC% of the prophage Escher_RCS47 with its chromosome and plasmid demonstrated the HGT of the prophage. However, prophage Escher_RCS47 located on plasmids seemed to be adapted to its respective genetic region, as indicated by GC%, suggesting the possibility of ancient prophage transfer events.

To further demonstrate this phenomenon, we next compared two other prophages with higher numbers on chromosomes and plasmids (Appendix A), and a similar conclusion was obtained. The average GC% of all 856 Escher_500465_1 prophages integrated on the chromosome was 55.2%, while the average bacterial chromosome GC% was 58.6% (Appendix A). The average GC% of all 536 Salmon_SEN34 prophages integrated on the chromosome was 54.1%, while the average bacterial chromosome GC% was 58.5% (Appendix A). The average GC% of all 1005 Salmon_SJ46 prophages integrated on plasmid was 54.0%, while the average bacterial plasmid GC% was 53.1% (Appendix A). The average GC% of all 360 Escher_SH2026Stx1 prophages integrated on plasmids was 53.8%, while the average bacterial plasmid GC% was 51.9% (Appendix A). This suggested that the prophages on the plasmids appeared to have adapted well to the respective genomic regions in which they were located and that there may be differences in the integration characteristics of prophages on chromosomes and plasmids.

## 3. Discussion

Horizontal gene transfer (HGT) is an important force in the evolution of bacteria, and these events often involve mobile genetic elements carrying genes related to adaptive strategies to promote dissemination [17]. As the driving force of HGT, prophages have been associated with the diversity and evolution of bacteria and may strongly affect the environmental adaptability of bacteria and their resistance to antibiotics. The prophages identified in some strains also appeared in other strains [18], providing evidence for horizontal gene transfer of prophages between strains. At present, there have been some articles on the integration of prophages in the genomes of different bacterial strains, and their characteristics and prevalences have been described [19,20,21,22,23]. In this study, we reported the prevalence of prophages in *K. pneumoniae*, analyzed different kinds of prophages integrated in chromosomes and plasmids, compared the relationship between strain types and prophage types, explored the relationship between the GC content of chromosomes or plasmids and the GC content of integrated prophages, and discussed their possible contribution to the pathogenicity and drug resistance of this opportunistic pathogen.

We found the presence of prophages in the chromosomes of 1437 strains of *K. pneumoniae*. Prophages were classified according to their degree of completeness, and the number of intact prophages on the chromosome was more than the sum of questionable and incomplete prophages. In contrast, on plasmids, prophages were predominantly incomplete. The high proportion of defective prophages on the plasmid may be related to the mobility and conjugation of the plasmid.

Some plasmids were identified as intact prophages, accounting for more than 90% or even approximately 100% of the total length of the plasmid. This suggested that some temperate phages actually exist in the form of extrachromosomal plasmids in the genome of *K. pneumoniae*. This was consistent with the plasmid–phage viewpoint mentioned in the study of phage N15 [24], pKO2 [25], and SSU5 [26]. In the present study, the analysis of intact prophage types close to the full length of the plasmid found that most of them were SSU5, accounting for 59.84% of these prophage. SSU5 was isolated from a *Salmonella enterica* strain [26] as a promising auxiliary component for phage cocktails [27]. This was followed by phiKO2 (31.89%) (Appendix A).

Prophages were found on all chromosomes, and approximately 61.86% of the plasmids had undergone integration by prophages. This also provided evidence that *K. pneumoniae* is highly lysogenic and may be related to its frequent exchange of genetic material [28]. In addition, high lysogenicity was also found in *Acinetobacter baumannii* [5], with a lysogenicity of 99.5%. Some strains seem to be more lysogenic than others, although the process associated with this change is still unknown. Touchon et al. [29] found that the minimum replication time and genome size were related to changes in lysogenicity. Fast-growing bacteria (with a minimum replication time of less than 2.5 h) showed greater lysogenicity than slow-growing bacteria. One explanation for this phenomenon is that fast-growing bacteria grow slowly under adverse environmental conditions [30]. In this environment, prophages are often thought to activate the lysogen cycle to protect their genomes while waiting for more favorable conditions to enter lytic reproduction [31]. The vast majority of *K. pneumoniae* in this study was isolated from the hospital environment and inpatients, which may be the reason for *K. pneumoniae* showing the integration of more prophages in the genome to adapt to this adverse environment. Our results were consistent with the findings of Costa et al. [5] in *Acinetobacter baumannii* and Touchon et al. [29] in mixed strains, i.e., the length of the chromosome was related to the number of integrated prophages. In addition, we also found that the length of the plasmid was related to the number of integrated prophages.

We detected VF and AMR genes in the strains. At the same time, a large number of VF and AMR genes were also found in prophages, but the proportion of these two genes encoded by prophages in different regions of the genome was inconsistent. Chromosomally integrated prophages had a much greater impact on strain virulence than plasmids, which may play a major role in the antibiotic resistance of strains. Our results will shed light on the important roles of prophages as reservoirs that transfer VF/AMR genes.

From our analysis of the GC content of the prophages integrated in the chromosomes and plasmids of *K. pneumoniae*, it was shown that the prophages within plasmids may have evolved in a different way than those integrated in chromosomes. Massive genetic acquisitions and losses are responsible for the evolution of prophages [8]. Further studies on the evolutionary mechanisms of prophages on chromosomes and plasmids will help us better understand the influence of prophages on host bacteria.

## 4. Materials and Methods

### 4.1. Sequence and Data Collection

The complete genomes of 1437 *K. pneumoniae* were retrieved from the National Center for Biotechnology Information (NCBI) nonredundant RefSeq database (https://www.ncbi.nlm.nih.gov/) (last accessed on 17 December 2022).

### 4.2. Typing of K. pneumoniae Strains

The associated types of MLST [32], K locus [33,34], and O locus [34,35] for each strain were identified using Pathogenwatch (https://pathogen.watch/, last accessed on 18 December 2022), a global platform for genomic surveillance.

### 4.3. Identification of Prophages in K. pneumoniae Strains

Whole genome sequences (including plasmids) of all the strains were used to identify and annotate the prophages by webserver PHASTER (PHAge Search Tool Enhanced Release, http://phaster.ca/, last accessed on 18 December 2022), a tool for rapid identification and annotation of prophage sequences within bacterial genomes [36]. According to the scoring criteria, the completeness of the prophage was scored by scanning the genes associated with the known phage and calculating the number of coding DNA sequences (CDSs) of the region where the putative prophages were located. On the basis of its completeness score, PHASTER separated the identified prophages into three types: intact (score > 90), questionable (score 70–90), and incomplete (score < 70). After removing the overlapping prophages that were predicted on the same chromosome or plasmid of the same strain, the remaining prophages were used for subsequent analysis.

### 4.4. Identification of Virulence Factors and Antibiotic Resistance Genes Carried by Prophages

A manually curated sequence set of 15,946 prophage genomes was used to search for putative virulence factor (VF) and antibiotic resistance (AMR) genes against the Virulence Factors of Pathogenic Bacteria Database (VFDB) (http://www.mgc.ac.cn/cgi-bin/VFs/v5/main.cgi, last accessed on 17 December 2022) [37] and the Comprehensive Antibiotic Resistance Database (CARD) (https://card.mcmaster.ca/, last accessed on 17 December 2022) [38], with a threshold of identity and coverage > 50% and e-value < 1 × 10^−5^.

### 4.5. Calculation of the GC Content of Prophages and the Whole Genome

The GC content of each prophage and the genetic region of the genome in which it was located was calculated using CodonW (https://sourceforge.net/projects/codonw/, last accessed on 18 December 2022).

### 4.6. Statistical Analysis

Statistical analysis of the data was performed using the independent samples Mann–Whitney U test for comparisons of two samples or the independent samples Kruskal–Wallis test for comparing multiple variables. The correlation between the integrated putative prophage numbers and the corresponding *K. pneumoniae* chromosome length was tested using the Spearman correlation coefficient test. The statistical inferences and hypothesis testing were generated using the software Origin 2021 and SPSS version 27 (IBM, Armonk, NY, USA), using a significance level of 95%.

## Figures and Tables

**Figure 1 ijms-24-09116-f001:**
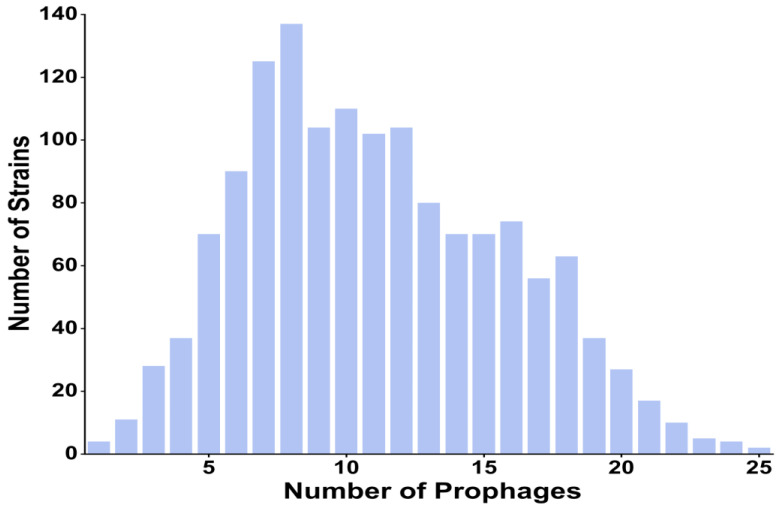
Bar graph depicting the distribution of total prophages (intact, questionable, and incomplete) in strains of *K. pneumoniae*.

**Figure 2 ijms-24-09116-f002:**
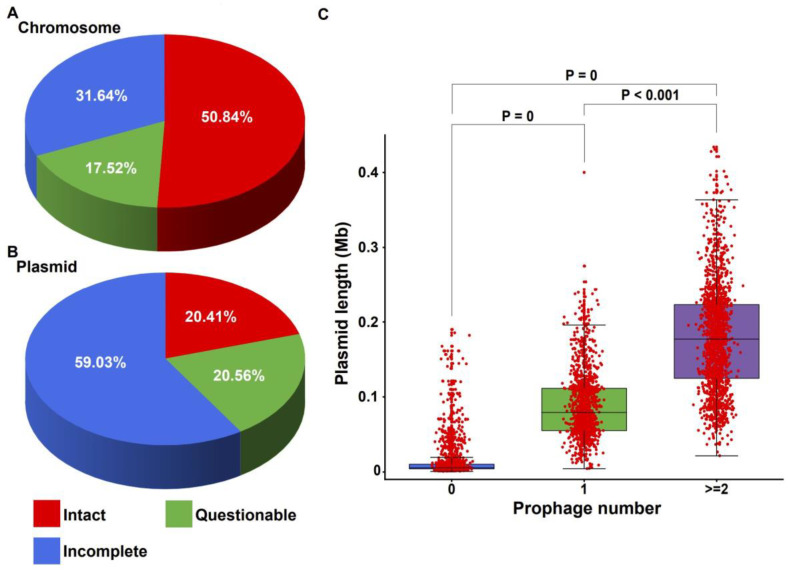
Distribution of prophages with different degrees of completeness on different genetic regions of the *K. pneumoniae* genome and correlation between plasmid length and the number of prophages it integrated. (**A**) The pie chart shows the respective proportions of intact, questionable, and incomplete prophages on the chromosomes. (**B**) The pie chart shows the respective proportions of intact, questionable, and incomplete prophages on the plasmids. The center of each part of the pie chart shows its respective scale. Red, intact prophages. Green, questionable prophages. Blue, incomplete prophages. (**C**) Box plot showing the relationship between bacterial plasmid length and the number of putative prophages per plasmid. The horizontal line at the center of the box represents the median. The bottom and top of the box represent the first and third quartiles, respectively. The external edges of the whiskers represent the lower and upper limit values, respectively. The red dots in the box plot represent individual plasmids. Significant differences (Kruskal–Wallis test) in *p* values are indicated.

**Figure 3 ijms-24-09116-f003:**
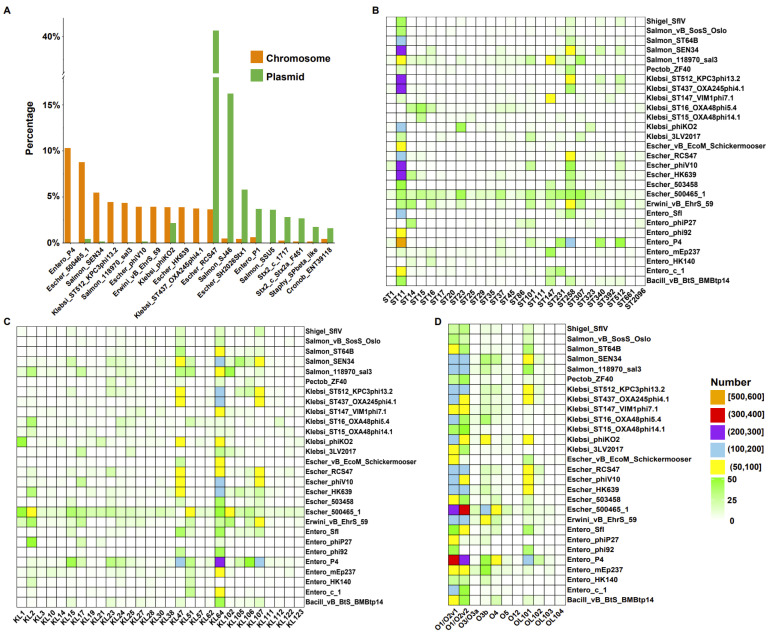
Comparison of prophages on chromosomes and plasmids and the relationship between strain types and types of prophages integrated with chromosomes. (**A**) Prophage types on chromosomes and plasmids and their respective proportions (top 10 for each). Horizontal coordinate, prophage type. Vertical coordinate, percentage. The vertical axis is truncated at 18–38. Orange, chromosome. Green, plasmid. (**B**) The relationship between sequence types and prophage types of chromosome integration. (**C**) The relationship between the type of K locus and the prophage types of chromosome integration. (**D**) The relationship between the types of LPS serotype (O locus) and the prophage types of chromosome integration. Sequence types, K-locus types, and O-locus types are shown along the X-axis, and prophage types are shown along the Y-axis. The color in the box indicates the number of prophages.

**Figure 4 ijms-24-09116-f004:**
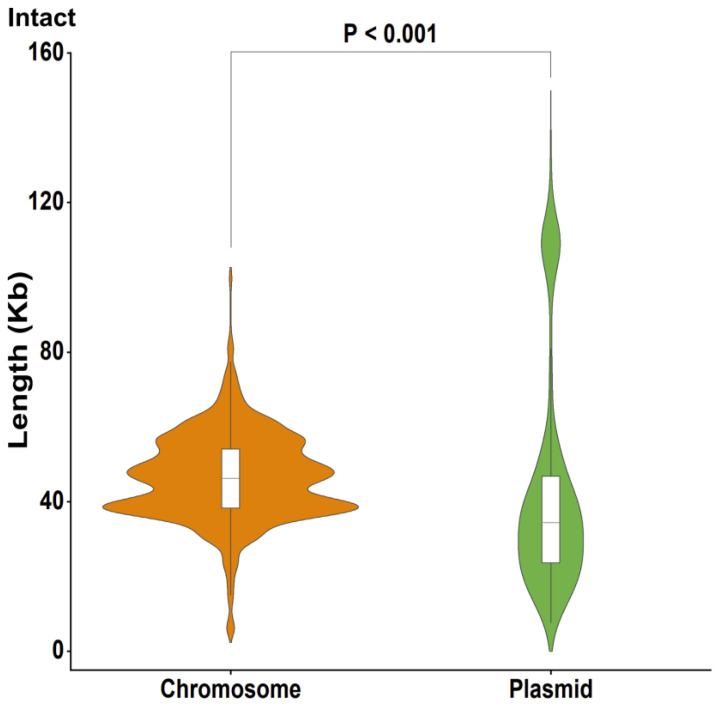
Violin plot showing the length of intact prophages integrated in chromosomes and plasmids. Violin plots show the number (width), median (centerline), interquartile range (hinges) and 1.5 times the interquartile range (adjacent lines). The sample sizes were as follows: chromosome, 4960; plasmid, 1264. Orange, chromosome. Green, plasmid. Significant differences (Mann–Whitney test) in *p* values are indicated.

**Figure 5 ijms-24-09116-f005:**
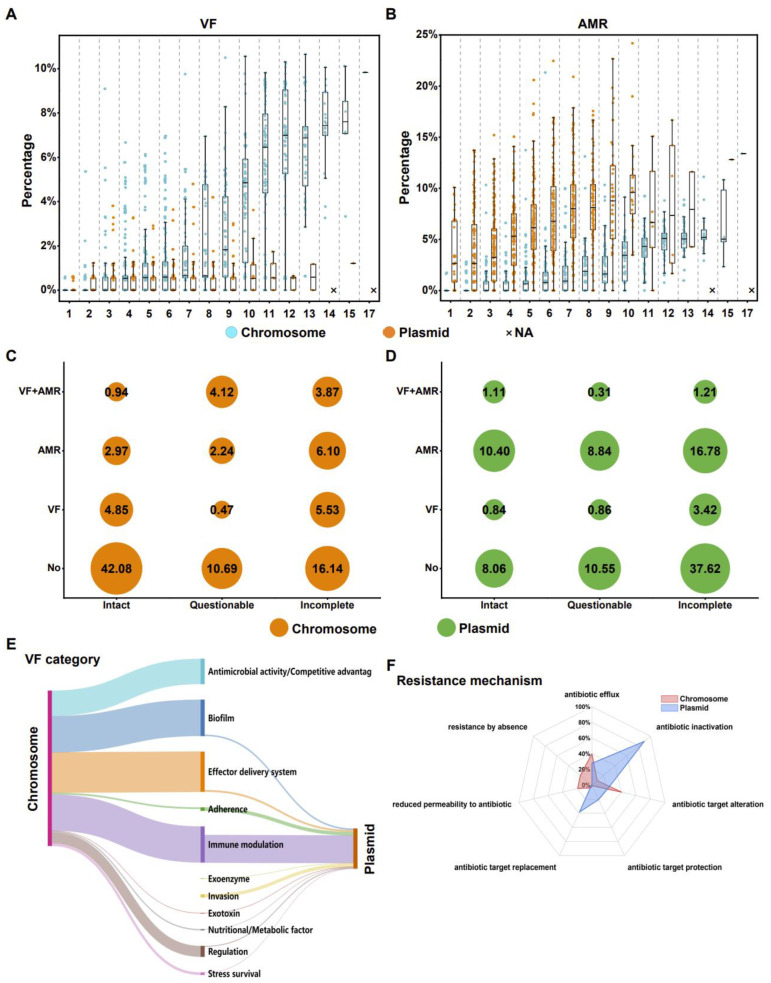
The effect of VF or AMR genes encoded by prophages on strains, and comparison of prophages encoding VF and AMR genes on chromosomes and plasmids. (**A**) Effect and proportional distribution of VF genes encoded by prophages on chromosomes and plasmids on strains. (**B**) Effect and proportional distribution of AMR genes encoded by prophages on chromosomes and plasmids on strains. Cyan, chromosome. Orange, plasmid. (**C**) The respective proportions of prophages with VF or AMR genes on chromosomes. (**D**) The respective proportions of prophages with VF or AMR genes on plasmids. The numbers in the solid circles indicate proportions. Orange, chromosome. Green, plasmid. (**E**) Functional comparison of virulence factors encoded by prophages on chromosomes and plasmids. Chromosomes and plasmids are on the left and right sides, respectively. In the middle are the functions of virulence factors. The width of the bands indicates the number of prophages with that function. (**F**) Comparison of antibiotic resistance mechanisms encoded by prophages on chromosomes and plasmids. Light-red shading, chromosomes. Light-blue shading, plasmid.

**Figure 6 ijms-24-09116-f006:**
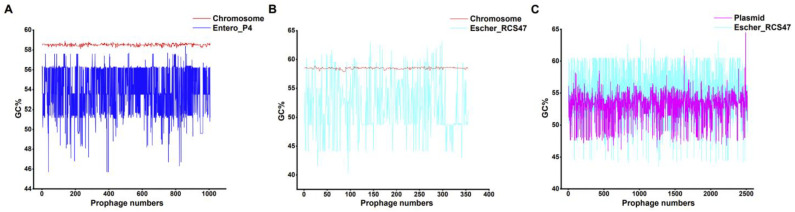
Comparison of the GC content of the prophage and the chromosome or plasmid on which it was located. (**A**) GC content of prophage Entero_P4 with its host chromosome. (**B**) GC content of prophage Escher_RCS47 with its host chromosome. (**C**) GC content of prophage Escher_RCS47 with its host plasmid. The X-axis indicates the number of prophages, and the Y-axis indicates the GC content of chromosomes, plasmids, or prophages. Red line, chromosome. Pink line, plasmid. Blue line, prophage Entero_P4. Cyan line, prophage Escher_RCS47.

## Data Availability

Not applicable.

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
