# Peer review of "Characterization and Diversity of Klebsiella pneumoniae Prophages"

_ijms, 2023, doi:10.3390/ijms24119116_

Round 1

Reviewer 1 Report

Journal: IJMS (ISSN 1422-0067)

Manuscript ID: ijms-2320228

Type: Article

Title: Characterization and diversity of Klebsiella pneumoniae prophages

Authors: Fuqiang Kang, Zili Chai, Beiping Li, Mingda Hu, Zilong Yang, Xia Wang, Wenting Liu, Hongguang Ren, Yuan Jin * , Junjie Yue *

Section: Molecular Pathology, Diagnostics, and Therapeutics

Special Issue: Molecular Advances in Infectious Disease

Dear Authors,

The Manuscript is very well written and great amount of results have shown in studies in regards to importance of presence of prophages on respective chromosomes and plasmids in K. pneumonie strains.  K. pneumoniae is a opportunistic pathogen and considered as resistance against several antibiotics (e.g,fluoroquinolones, beta-lactams, penicillins, cephalosporins and carbapenems). This high rate of resistivity against several drugs is the serious challenge for scientist to treat Klebsiella infections in humans. The prevalence of multi-drug resistance (MDR) Klebsiella has increased exponentially in recent times and it is important  to find alternative treatment therapy to control Klebsiella infection. Klebsiella  has several Virulence factor (VF) and antibiotic resistance genes (AMR) which is carried by prophages present in Klebsiella strains on chromosome and plasmids. Authors have given good background information in introduction for Klebsiella prophages and it's importance. The intent of this study to understand the resistance ,virulence  pattern of Klebsiella which is constantly increasing due to large accessory genome. Phage therapy is the potential choice for fast emerging MDR K. penumonie strains. Authors have described presence of prophage on chromosome and plasmids and its effect on virulence and AMR gene. they have given overall good amount of data which shows prophage percentage, length, gene loci. It's overwhelming to see so much data and so many numbers appreciate their effort to find from gene database and decoding information to understand the pathogen mechanism, however, it is so many number mentioned in manuscript for mentioning presence of prophage, length and GC content. I would definitely recommend alternative therapy for such dangerous pathogen and it is fast emerging threat.

Major : It is hard to summarise the numbers mentioned in manuscript from their results. Graphs are great way to explain things however, after the final evolution there could have been great way to summarise the results detected in this study by incorporating final data in comparative table format in correspondence to  chromosome and Plasmids form Klebsiella.  Every time we have to go back find the numbers  (length, GC content, percentage, proportions, gene loci) mentioned in manuscript which is not easy.  It's really good comparative studies, however, it would be great  if simplified by putting all your results of all diagrams in one table or flow chart. Which can be useful for future reference too. Authors have done excellent observation however it is overwhelming data. I would advice to simplify their findings which will make their manuscript most important  for readers.

Minor: There are very small minor corrections which can be easily corrected during final submission.

Reviewer 2 Report

The study of Klebsiella pneumoniae is being done for more than a century, and its capability of transfer genetic material. You have done a work using mainly software, and i'll consider ok. There is no conclusion so i assume this is a introduction of a thesis or preparing another study.

Keep up good work.

Reviewer 3 Report

Generally the manuscript is well written.

Line 74: please explain "putative prophages were not identified in 38.14%.". 38.14% of what? Plasmids? Also what are putative prohages?

Line 74: probably you mean 50.84% intact prophages.

Line 83: please explain the defective prophages meaning in parentheses instead in commas

 Line 106: The ohrase don't have logic. Please revise.

Line 222: Please rephrase
